# Mapping Chilean clinical research: a protocol for a scoping review and multiple evidence gap maps

Javier Bracchiglione [ORCID],[1] Nicolás Meza [ORCID],[1] Juan Victor Ariel Franco,[2] Camila Micaela Escobar Liquitay [ORCID],[3] Sergio R Munoz [ORCID],[4] Gerard Urrutia,[5] Eva Madrid [ORCID][1]

[1]Interdisciplinary Centre for Health Studies (CIESAL), Universidad de Valparaíso, Valparaíso, Chile
[2]Institute of General Practice, Heinrich-Heine-University Düsseldorf, Düsseldorf, Germany
[3]Research Department, Instituto Universitario Hospital Italiano de Buenos Aires, Buenos Aires, Argentina
[4]Department of Public Health–CIGES, Universidad de La Frontera, Temuco, Chile
[5]Biomedical Research Institute Sant Pau (IIB Sant Pau), CIBER Epidemiología y Salud Pública (CIBERESP), Iberoamerican Cochrane Centre, Barcelona, Spain

**Correspondence to**
Dr Nicolás Meza;
nicolas.meza@uv.cl

## ABSTRACT

**Introduction** Clinical research broadly aims to influence decision-making in order to promote appropriate healthcare. Funding agencies should prioritise research projects according to needed research topics, methodological and cost-effectiveness considerations, and expected social value. In Chile, there is no local diagnosis regarding recent clinical research that might inform prioritisation for future research funding. This research aims to comprehensively identify and classify Chilean health research studies, elaborating evidence gap maps for the most burdensome local conditions.

**Methods and analysis** We will search in electronic databases (MEDLINE, Embase, PsycINFO, CINAHL, LILACS and WoS) and perform hand searches to retrieve, identify and classify health research studies conducted in Chile or by authors whose affiliations are based in Chile, from 2000 onwards. We will elaborate evidence matrices for the 20 conditions with the highest burden in Chile (according to the Global Burden of Disease 2019) selected from those defined under the General Regime of the Health Guarantees Act. To elaborate the evidence gap maps, we will consider prioritised interventions and core outcome sets. To identify knowledge gaps and estimate redundant research, we will contrast these gap maps with the available international evidence of high or moderate certainty of evidence, for each specific clinical question. For this purpose, we will search systematic reviews using the Grading of Recommendations, Assessment, Development and Evaluation (GRADE) approach.

**Ethics and dissemination** No ethical approval is required to conduct this project. We will submit our results in both peer-reviewed journals and scientific conferences. We will aim to disseminate our findings through different academic platforms, social media, local press, among others. The final results will be communicated to local funding agencies and government stakeholders.

**Discussion** We aim to provide an accurate and up-to-date picture of the research gaps—to be filled by new future findings—and the identification of redundant research, which will constitute relevant information for local decision-makers.

## INTRODUCTION

Primary clinical research and synthesised evidence are the basis for knowledge

## STRENGTHS AND LIMITATIONS OF THIS STUDY

⇒ The protocol for this scoping review includes a comprehensive electronic and hand search strategy.
⇒ We will be able to accurately elucidate local evidence gaps by first constructing evidence grids and then populating them with the results of our search.
⇒ Considering that we used the estimations from the Global Burden of Disease 2019 for prioritising the most important diseases in Chile, we might be neglecting some important clinical entities, such as multicomponent diseases.
⇒ We will develop gap maps only for the 20 conditions with the highest burden in Chile.

translation, decision support and implementation in clinical settings, which should ultimately guide the generation of primary research (on prevention, aetiology, diagnosis, treatment and prognosis of any health condition or disease).[1 2]

Good quality clinical research, which must be set on priority questions,[3 4] is essential for elucidating the best approaches to solve health challenges;[5] aiming at informing decision-making to make healthcare more effective, less harmful and less expensive.[6] However, much concern exists about poor design and deficient conduct and reporting,[7] not only because of methodological issues or its effects on scientific advancement but by its practical implications, at either individual or population level.[8] Glasziou and Chalmers[8] estimated that 85% of the investments for clinical research end up being avoidable research waste, which may be derived from any step along the research process and its application (ie, disregarding regulation, governance and management over clinical research development).[3 9–12] Stakeholders should have accurate and up-to-date pictures of evidence so that resources are used in the best possible way.[13] Nevertheless, the use of rigorous evidence for both clinical practice

and health policy-making is still limited.[14–16] The limited budgets make prioritisation a mandatory step for funding agencies; which should be done mainly considering the needed research topics, cost-effectiveness and expected social value.[17 18]

Although some frameworks have been proposed to evaluate research applications,[19] the focus on academic background and scientific productivity during the project assessment seem to be predominant.[20] In practice, organisations summon expert panels to judge according to the afore-mentioned technical and academic merit measures, with heterogeneous considerations about cost, research gaps, social value or local impact.[20]

The lack of a prioritised agenda in developing countries may generate a source of inequity, by letting third-party interests (ie, global research funding, or the pharmaceutical industry from high-income countries) inhabit regional research planning,[21–23] in a context of poor governance in this matter.[12 24] The strategic planning of state funding allocation in most countries can still improve the identification of local evidence gaps to be filled by new research, avoiding (and not funding) redundant research. In Chile, clinical research is funded through different government grants, in addition to other types of funding (private agencies, competitive funding, pharmaceutical or medical device industry sponsoring, international grants, etc),[25] but there is no comprehensive local diagnosis of recent clinical research that could inform prioritisation for future research funding.

This is a protocol for a government-funded project with the following objectives:

1. To identify and classify health research studies conducted in Chile or by authors whose affiliations are based in Chile.
2. To elaborate evidence maps for relevant health conditions in Chile, considering prioritised interventions and core outcome sets.
3. To populate the maps with the primary and secondary evidence and their risk of bias and to identify knowledge gaps and redundant research incorporating international evidence.

## METHODS AND ANALYSIS

This section describes the methods for each of the objectives.

### Objective 1: identification and classification of Chilean clinical research

We will conduct a scoping review that will be reported following the Preferred Reporting Items for Systematic Reviews and Meta-Analyses extension for scoping reviews (PRISMA-ScR).[26] For this protocol, we followed the guidance from the PRISMA-P extension for protocols.[27]

### Eligibility criteria

We will consider any clinical study design, either conducted in Chile or at least one affiliation based in Chile. We define clinical studies as those focused on a clinical health topic describing, measuring or exploring a health-related outcome in humans. Considering that our focus in the following objectives will be the highest priority conditions of the last Chilean health reform, we will consider studies published from 2000 onwards.[28]

We will consider any descriptive, observational or experimental primary study designs, including case reports, case series, cross-sectional studies, case–control studies, cohort studies, quasiexperimental studies, diagnostic or prognostic studies, mixed methods studies, and clinical trials. We will also include different formats of evidence synthesis, including systematic reviews, scoping reviews, evidence maps and overviews.[29] We will exclude narrative reviews, editorials, correspondence, letters to the editor, opinion articles, conference proceedings, study protocols and preprint reports without peer review. We will also exclude studies whose observation units are biological samples, economic evaluations, modelling studies, validation of instruments and ecological studies.

### Search methods

#### Electronic search strategy

We will perform a comprehensive search in electronic databases, without restriction of language or publication status.

We will search the following databases:

1. MEDLINE via Ovid SP.
2. Embase via Elsevier.
3. PsycINFO via ProQuest SP.
4. Cumulative Index to Nursing and Allied Health Literature (CINAHL) via EBSCOhost.
5. Latin American and Caribbean Literature in Health Sciences (LILACS).
6. Web of Science (WoS) via Clarivate.

Online supplemental appendix A provides details of the electronic search strategy for MEDLINE via Ovid. For our MEDLINE search, we added a highly sensitive filter to identify randomised trials developed by the Cochrane Collaboration[30] and a validated search filter to retrieve systematic reviews developed by the Scottish Intercollegiate Guidelines Network.[31] The strategy proposed was peer-reviewed by another information specialist prior to implementation using the Peer Review of Electronic Search Strategies (PRESS) checklist.[32]

#### Hand search strategy

We will conduct complimentary hand searches with the same criteria reported in the electronic search. We identified seven information resources considered relevant to identify all the evidence that has not been indexed in the databases described in the electronic search and additional four information resources necessary to retrieve trials, reviews or other types of evidence,

published peer reviewed research, that meet the eligibility criteria (see online supplemental appendix B for further details).

## Study selection

Two reviewers will independently perform a title and abstract screening. A third reviewer will solve any discrepancies. We will retrieve the full text of each study applying our selection criteria to determine its final inclusion. Afterwards, two reviewers will screen references by full text, solving discrepancies by a third reviewer. For this process, we will use the Covidence platform.[33] We will present a PRISMA 2020 flow diagram showing the process of study selection.[34]

## Data extraction

Two reviewers will extract the following data from the included studies:
► Bibliographic data: full citation including the list of authors, journal and date of publication.
► Type of evidence: primary studies or secondary studies.
► Study design: case reports, case series, cross-sectional studies, case–control studies, cohorts, quasiexperimental studies, diagnostic studies, prognostic studies, randomised trials, systematic reviews and other forms of synthesised evidence.
► Area of study: we will characterise the retrieved articles by discipline and area of study used by the Organisation for Economic Co-operation and Development in their category scheme.[35]
► Location of the study: Chile, other country, multicentric study.
► Authors and authorships: affiliation (based in Chilean institution or not), gender and type of authorship (main author, last author, corresponding author, working groups authorships, among others).
► Diseases or health conditions addressed by each study according to the taxonomy developed by the Global Burden of Disease.[36]
► Funding: we will classify the type of funding of each article, public or private, competitive funding, pharmaceutical or medical device industry sponsoring, or international grants.
► Conflicts of interest: we will extract the authors' statements from each report, considering descriptions for each author if available.

We will enter the data into a data extraction form (based in Google Sheets, Google).

## Summary of data

We do not intend to perform a risk-of-bias assessment at this stage (see Objective 3). We will summarise the findings of each category (ie, proportion of clinical trials, proportion of studies for each condition). Moreover, we will illustrate the trends for each study design and disease category.

## Objective 2: elaboration of evidence matrices for relevant health conditions in Chile, considering prioritised interventions and core outcome sets

We will elaborate evidence matrices for prioritised health conditions selected from those defined under the General Regime of the Health Guarantees Act (Garantías Explícitas en Salud (GES)) that was stated in 2004 in Chile. This Regime commanded public and private health providers to guarantee access, opportunity, quality and financial protection for the most relevant programmes, diseases or conditions.[37] This Regime contemplates 85 programmes, diseases and conditions that have been selected considering the health situation of the population, the effectiveness of the interventions, their contribution to the extension or quality of life and, when possible, their cost-effectiveness.[38]

We extracted the list of health conditions and excluded those defined as programmes (eg, 'orthosis (or technical help) for people aged 65 years and older' which aims to improve independence in the elderly). These conditions were initially prioritised by the Chilean government based on a local study of the GBD in 2007.[39] In consultation with the health ministry, we decided to prioritise the conditions based on the 2019 report by the GBD initiative as it is a more up-to-date resource for decision-making.[40] GBD is a consortium of more than 3600 researchers who collect, analyse data and provide a tool to visualise the burden and health loss of people due to hundreds of health conditions.[40] We filtered the GBD database by year (2019), location (Chile), context (cause), measure (disability-adjusted life years) and metric (number). We distinguished causes by age groups, and used data from all causes and both sexes. We excluded GES conditions that did not match precisely enough with a GBD cause, according to the consensus of the authors. We provide the details and judgements regarding the reasons for exclusions for each GES problem in online supplemental appendix C. Then, we selected the 20 conditions with the highest burden in Chile, among those included in GES. This selection closely matched the local study on burden of disease conducted in 2007, which informed decision-making during the constitution of GES.[39] See table 1 for the list of the top 20 conditions identified in this process.

We will then conduct the following steps to complete the matrices that will be populated with evidence in the following objective.

### Step 1: identifying the main outcomes

The definition of main outcomes for each health condition is a complex and difficult task, considering that reported outcomes are prone to bias, and that many studies tend to report outcomes with positive or statistically significant findings.[41]

For that reason, we will consider agreed standardised sets of outcomes, known as core outcome sets (COS) for identifying the main outcomes. The COS represents a non-restrictive minimum set of outcomes to be assessed and reported in studies for every health condition, and

**Table 1**  Prioritised conditions from the General Regime of the Health Guarantees Act (Garantías Explícitas en Salud (GES)) according to the Global Burden of Disease (GBD) 2019 in Chile, in decreasing order by disability-adjusted life years (DALYs)

| GES condition | Matching GBD cause (age group) | GBD 2019 DALYs, number (95% CI, upper to lower) |
| --- | --- | --- |
| Myocardial infarction | Ischaemic heart disease (all ages) | 214 819.57 (227 068.32 to 0) |
| Type 2 diabetes | Diabetes mellitus type 2 (all ages) | 170 569.88 (213 283.00 to 0.19) |
| Depression in people aged 15 years and over | Depressive disorders (20 plus) | 121 414.38 (169 061.49 to 2.23E−06) |
| Chronic kidney disease stage 4 and 6 | Chronic kidney disease (all ages) | 101 733.72 (110 880.12 to 208.53) |
| Ischaemic stroke in people aged 15 years and over | Ischaemic stroke (20 plus) | 100 161.76 (109 460.56 to 94.57) |
| Stomach cancer | Stomach cancer (all ages) | 85 929.07 (91 647.46 to 1.26) |
| Lung cancer | Tracheal, bronchus and lung cancer (all ages) | 83 674.49 (88 943.91 to 0.0002) |
| Chronic obstructive pulmonary disease (outpatient management) | Chronic obstructive pulmonary disease (all ages) | 83 190.76 (96 438.34 to 0.0002) |
| Alzheimer's disease and other dementias | Alzheimer's disease and other dementias (all ages) | 72 825.97 (154 958.48 to 4.11) |
| Colorectal cancer in people aged 15 years and over | Colon and rectum cancer (20 plus) | 70 756.65 (75 584.77 to 1.66) |
| Hip and/or knee osteoarthritis, mild or moderate, in people aged 55 years and over (medical management) | Osteoarthritis (55 plus) | 55 576.88 (112 169.90 to 1.85) |
| Prevention of preterm birth | Neonatal preterm birth (all ages) | 51 479.73 (63 435.89 to 0.16) |
| Chronic hepatitis C | Cirrhosis and other chronic liver diseases due to hepatitis C (all ages) | 50 969.21 (65 323.67 to 0.0005) |
| Breast cancer in people aged 15 years and over | Breast cancer (20 plus) | 47 006.54 (51 554.51 to 16.13) |
| Prostate cancer in people aged 15 years and over | Prostate cancer (20 plus) | 45 854.87 (54 664.48 to 2.60) |
| Schizophrenia | Schizophrenia (all ages) | 42 501.32 (56 297.82 to 0.24) |
| Bipolar disorder in people aged 15 years and over | Bipolar disorder (20 plus) | 38 096.12 (58 487.76 to 8.43E−06) |
| Community-acquired pneumonia in people aged 65 years and over (outpatient management) | Lower respiratory infections (65–89 years) | 34 844.45 (38 946.70 to 1.10) |
| Asthma in people aged 15 years and over | Asthma (20 plus) | 32 694.24 (45 713.86 to 0.12) |
| Secondary subarachnoid haemorrhage to rupture of brain aneurysms | Subarachnoid haemorrhage (all ages) | 28 635.44 (31 612.39 to 1089.41) |

they are permanently updated and revised by the Core Outcome Measures in Effectiveness Trials (COMET) initiative (http://www.comet-initiative.org/). The COS includes outcomes that are most relevant to clinicians, decision-makers, patients and carers. If COS are not available for an individual condition, we will build a set of outcomes based on consensus with the input of multidisciplinary experts and patients, considering the existing outcomes embedded in current GES clinical practice guidelines.

### Step 2: identifying the interventions
We will define the interventions for the rows of the matrix for each health condition considering:
► Local Clinical Practice Guidelines: we will extract the main interventions from the clinical recommendations

in existing guidelines conducted by the Chilean Health Ministry.
► Those prioritised by Cochrane Review Groups and Networks: we will contact Cochrane Groups and Networks' authors in order to gather a list of priority interventions for each health condition, according to their consideration.
► Those identified by the regulatory agency: we will review the authorisations of the Institute of Public Health in Chile as the main regulatory agency in charge of the permits for clinical trials, in order to classify the authorised pharmacological interventions for each health condition.

### Step 3: building up the matrices
We will create evidence gap maps for each included condition. These evidence gap maps will be framed on grids or matrices that will consider the relevant interventions (defined in Step 3) in the rows, and the main outcomes (defined in Step 2) in the columns. These grids will be populated in each intersection with the included studies, as we detail in Objective 3 (see Objective 3 below). We will use evimappr,[42] an R[43] package for producing bubble plots, which provides an interactive display for visualising the evidence gap maps.

### Objective 3: to populate the maps with local primary and secondary evidence and their risk of bias and to identify knowledge gaps and redundant research incorporating international evidence
After the completion of the tasks proposed in Objective 1, we will select all the studies that might provide evidence for decision-making regarding interventions, to populate each of the matrices elaborated as described in Objective 2. Whenever an evidence matrix is populated with studies for a certain outcome and intervention, we will refer to it as an evidence map.

### Step 1: identifying the main study designs
From the complete set of research articles included in Objective 1, we will separate all those studies with a design relevant for decision-making regarding interventions:
1. Randomised controlled trials.
2. Non-randomised primary studies, including controlled clinical trials, quasiexperimental designs (eg, interrupted time series, controlled before–after studies) and observational studies (cohort studies, case–control studies, analytical cross-sectional studies).
3. Synthesised evidence relevant to interventions (systematic reviews, overviews).

Descriptive studies, qualitative or mixed methods studies, diagnostic accuracy studies, narrative reviews, scoping reviews and evidence maps will not be considered for this objective as they do not provide evidence supporting the efficacy or effectiveness of interventions.

### Step 2: risk-of-bias assessment
To assess risk of bias of the included studies, we will use standardised tools for each methodological design:

(1) Cochrane Risk-of-Bias tool for randomised clinical trials,[44] (2) Risk Of Bias In Non-randomised Studies - of Interventions (ROBINS-I) for non-randomised primary studies[45] and (3) Risk of Bias Assessment Tool for Systematic Reviews (ROBIS) for systematic reviews.[46] We will skip this step in the case of overviews, since currently there is no validated tool for assessing risk of bias in this specific methodological design.

### Step 3: populating evidence gap maps
We will populate each node of the previously elaborated grids or matrices with the selected studies, considering the intervention and outcome on that node. We will represent the populated nodes with visual symbols (bubbles), with different colours according to the type of study and risk of bias, and different sizes according to the number of studies available. Once a matrix is populated with Chilean studies, we will refer to it as a 'local evidence map'.

### Step 4: developing global certainty of the evidence gap maps
We will develop a second map for each prioritised condition populated with rigorous international research (defined as the availability of high or moderate certainty of evidence), for each specific clinical question, according to the Grading of Recommendations, Assessment, Development and Evaluation (GRADE) approach,[47] that we will refer to as a 'global evidence map'. For this purpose, we will conduct a new search aimed at identifying systematic reviews using the GRADE approach, using appropriate filters.[32] We will populate these maps with the degree of certainty of evidence, so each node will be classified as 'high certainty' or 'moderate certainty' (reflecting that mostly there is enough evidence); and 'low certainty', 'very low certainty', or 'no evidence' (reflecting that more evidence is needed).

### Step 5: contrasting global certainty of the evidence maps with local maps
Once Step 4 is concluded, we will have two maps for each condition: (1) a local evidence map (populated with local Chilean research) and 2) a global certainty of the evidence map (populated with certainties of evidence from international synthesised evidence). In order to identify knowledge gaps and estimate redundant research, we will contrast these two maps. Once we cross these two maps, we will classify each populated node of the local map as 'adequate research' or 'redundant research', while each empty node will be classified as 'evidence gap', according to the criteria defined in table 2.

We will consider the absence of local evidence as a gap. If this gap is established in the context of low or very low certainty of evidence in the global map, or if there is no global evidence, we will consider it as a true gap. If the gap is established in the context of high or moderate certainty of evidence in the global gap, we will consider it in general as a false evidence gap.

If there is local research in the context of a low, very low certainty or no evidence in the global map, we will

**Table 2** Possible scenarios in the evidence gap maps

| | Global evidence map: very low to low certainty of the evidence (or no evidence) | Global evidence map: moderate to high certainty of the evidence |
|---|---|---|
| Local evidence map No local evidence | True evidence gap | False evidence gap (unless local conditions require context-specific research) |
| Local evidence map Available local evidence | Adequate research | Adequate research (if local evidence was published before the review) Redundant research (if local evidence was published after the review) |

consider the local research as adequate. If there is local research in the context of a high or moderate certainty of evidence in the global map, we will consider the research as adequate if it was published before the review that informed the certainty of evidence, or as redundant research if it was published after this review.

### Patient and public involvement

No patients were involved in the development of this protocol.

### Ethics and dissemination

No ethical approval is required to conduct this project. Our findings will be submitted to peer-reviewed journals and scientific conferences. To amplify the impact of our work and to accomplish knowledge translation objectives, we will disseminate the results through different platforms, including academic social media, blogs, local press, among others. At the same time, we will aim to communicate the final results to local funding agencies and government stakeholders, in order to inform the elaboration of future national research agendas.

### DISCUSSION

After the completion of this proposed study, we will deliver an overall picture of the clinical research conducted in Chile since 2000, and we will elaborate evidence gap maps for the main health conditions guaranteed in Chile, with a detailed description of the most relevant interventions and outcomes and the type of clinical research in each node.

We will also study the evidence gaps for the 10 health conditions with the highest burden in Chile and compare them with the available research globally; as a way to establish the true gaps of clinical research, and the amount of redundant findings addressed in Chile or by authors affiliated to Chilean institutions.

A limitation of our protocol might be the use of the GBD 2019 estimations to prioritise the most important diseases or conditions in Chile, according to GES. As GES is based on policy and local definitions, and not totally on an epidemiological rationale, our conceptualisation may neglect some important clinical entities (mainly multicomponent diseases). Nevertheless, we have added online supplemental appendices with the detailed pairing process, the burden of disease and the excluded health topics (and reason of exclusion). To operationalise our procedures, we also narrowed the scope of the gap maps development (Objective 3) to clinical research centred on efficacy or effectiveness, excluding

descriptive studies, qualitative or mixed methods studies and diagnostic accuracy studies, among others. Furthermore, we will limit the evidence maps only to health interventions in prioritised outcomes, and then, by focusing on the 20 conditions with the highest burden in Chile, we will not address the whole extent of the research in locally prioritised health conditions.

A main strength of our protocol is the comprehensiveness of our search strategy. The development of a peer-reviewed search strategy in major electronic databases complemented with an exhaustive hand search will allow us to probably identify the whole body of clinical evidence conducted locally. The hand search is crucial and constitutes an important source of information, considering that research in the Latin American Region might not be adequately indexed in the electronic databases. The process of development of evidence maps constitute another major strength of our proposed research. We will first develop the evidence grids identifying main outcomes and interventions independently of the results of our search and screening process (Objective 2), and we will populate those grids afterwards with the results of our selection process (Objective 3). Besides identifying the research being conducted in Chile, with this approach, we will be able to accurately elucidate the evidence gaps, since it is possible that particular interventions in the grid remain unpopulated, as the construction of the grid will not be guided by the results of our search.

This accurate and up-to-date picture of the research gaps—to be filled by new future findings—and the identification of redundant research will constitute relevant information for local decision-makers and, at the same time, an interesting methodological proposal to visualise clinical research trends and gaps in other countries or regions.

**Acknowledgements** We thank Ivan Solà from the Iberoamerican Cochrane Centre—Sant Pau Biomedical Research Institute (IIB Sant Pau)—CIBERESP (Barcelona, Spain) for his valuable support in drafting and refining our search strategy.

**Contributors** The study concept was developed by JB, NM, JVAF and EM. The manuscript of the protocol was drafted by JB, NM, JVAF, CMEL and EM. SRM and GU developed and provided feedback for all sections of the review protocol and approved the final manuscript. The search strategy was developed by CMEL and JB. Study selection, data extraction, quality assessment and gap maps elaboration will be performed by all the authors. All authors reviewed and approved the final version of the manuscript.

**Funding** FONDECYT Grant 1212037 from the Chilean National Agency of Research and Development (ANID).

**Disclaimer** The funding agency had no involvement in the conception, development, drafting or approval of this manuscript.

**Competing interests** All the authors have conducted clinical research in Chile or associated with Chilean researchers or institutions. Nevertheless, decisions about inclusion, data extraction and quality assessment of these studies will be conducted by independent researchers within our team. We declare no other conflict of interests.

**Patient and public involvement** Patients and/or the public were not involved in the design, or conduct, or reporting, or dissemination plans of this research.

**Patient consent for publication** Not applicable.

**Provenance and peer review** Not commissioned; externally peer reviewed.

**ORCID iDs**
Javier Bracchiglione http://orcid.org/0000-0001-8738-2184
Nicolás Meza http://orcid.org/0000-0001-9505-0358
Camila Micaela Escobar Liquitay http://orcid.org/0000-0002-2903-6870
Sergio R Munoz http://orcid.org/0000-0001-8383-6599
Eva Madrid http://orcid.org/0000-0002-8095-5549

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
