## [Reviewer comments · BMJ Open]

ARTICLE DETAILS

TITLE (PROVISIONAL)	Mapping Chilean clinical research: a protocol for a scoping review and multiple evidence gap maps
AUTHORS	Bracchiglione, Javier; Meza, Nicolás; Franco, Juan Victor Ariel; Escobar Liquitay, Camila Micaela; Munoz, Sergio; Urrutia, Gerard; Madrid, Eva

VERSION 1 – REVIEW

REVIEWER	Nyanchoka, Linda Université Paris Descartes
REVIEW RETURNED	27-Oct-2021

GENERAL COMMENTS	This is a very important and interesting protocol. The authors clearly justify the importance of gap identification and research prioritization to improve the conduct, reporting and use of clinical research. I wish the others a successful study and look forward to their actual study and findings. I would like to mainly comment on the methodology proposed. I would suggest the authors provide reflections on the strengths and anticipated limitations of the planned study methodologies to identify gaps and propose priorities. Moreover describe some of their attempts to address the limitations. Finally, due to the innovative nature of their study, it would be useful to provide future considerations that would optimize their study that are not in place, for example the role of technology in gap identification and research prioritization. Overall, I look forward to their study and study findings. All the best.
--

REVIEWER	Gustafsson, Lars Karolinska Institutet, Department of Laboratory medicine
REVIEW RETURNED	13-Dec-2021

GENERAL COMMENTS	The authors are presenting a protocol for a review of clinical Chilean research publications from 2000 linked to an analysis of prioritized health research. It is a welcomed study but the submitted study protocol manuscript needs to be revised and clarified concerning design, used methods and on the linkage to prioritized health conditions in Chile. MAJOR COMMENTS 1. I have a hard time to understand that the listed prioritized health conditions in Table I represent the ESSENTIAL HEALTH NEEDS in Chile. Chile represents to me a middle income country that certainly has to be stuck by preventable disorders like alcohol and drug
---

	abuse, road accidents and nutritional disorders among children and need to handle multidiseased patients. The authors need to convince me and the readers that the list of health conditions do represent the 20 most important according to burden of disease. Table I represents a focus on single organ based diseases that generally do not represented the highest burden of disease in most countries. 2. It is not clear what type of publications are meant to be included in the review. Please, clarify since it is not clear why mixed-methods projects and laboratory studies are not included (page 5). New laboratory diagnostic methods might be key both in direct care and in preventive studies. The authors list an authoritative number of sties to be searched for Chilean publications (top page 8) and that is adequate. However, I do not understand the criteria for hand search strategy in a number of sites including various repositories. It might be relevant to include a hand-search but in that case it IS NOT RECOMMENDED TO INCLUDE PUBLICATIONS FOR ANALYSIS UNLESS THEY HAVE UNDERGONE PEER-REVIEW. The authors are recommended to include a flow-chart Table summarizing the different types of publication sites to be searched and criteria for inclusion and exclusion of papers in the final analysis. 3. The authors should revise parameters to be reviewed in each paper and consider to list if the authors REPORT CONFLICT OF INTERESTS. The authors underestimate how bias and conflict of interests can affect results of various clinical studies. Recently, this issue was addressed by an excellent review in British Medical Journal related to medical product industry (Chimonas S, Mamoor M, Zimbalist SA, Barrow B, Bach PB, Korenstein D. Mapping conflict of interests: scoping review. BMJ 202;375:e066576.) It might be adequate to include references addressing the issues on how conflict of interests bias selection of research areas and study designs. 4. This study protocol is designed based on the concept that it is easy to define what are the needs of health research. In fact, definition of the health and research needs need to be open for discussion eventhough if they are defined of public bodies like ministries. It seems as if the authors underestimate the power of curiosity driven basic as well as clinical research. Please, address in the background how you consider the role of basic and clinical research to address major health and medical needs in Chile. It might be a drawback for Chilean health systems to thrive in a university and university hospital with adequate resources for curiosity driven basic and clinical research. 5. English grammar and style need to carefully reviewed. MINOR ISSUES 6. The listed key words on the front page do not seem adequate for the title of the paper and do not agree with the key words listed on page 2. 7. Strengths and limitations of the study-page 2-3 should be revised and not only describe what the study is about.
--	--

	8. A number of terms are used in the article. Consider to include a list of definitions. Clarify also what you mean of waste of research. You cite a number articles on this subject but the term waste is not well-defined. 9. Please clarify hwo the study is funded and the role of funding agencies. 10. It is remarkable that the authors do not state if you have any conflicts of interests. 11. I can not find that the authors have completed the PRISMA protocol as requested by the journal.
--	---

VERSION 1 – AUTHOR RESPONSE

Reviewer 1: Ms. Linda Nyanchoka, Université Paris Descartes, University of Liverpool.

Reviewer's feedback	Authors' response
This is a very important and interesting protocol. The authors clearly justify the importance of gap identification and research prioritization to improve the conduct , reporting and use of clinical research. I wish the authors a successful study and look forward to there actual study and findings. I would like to mainly comment on the methodology proposed. I would suggest the authors provide reflections on the strengths and anticipated limitations of the planned study methodologies to identify gaps and propose priorities. Moreover describe some of their attempts to address the limitations.	We are very grateful about this comment. We mainly believe that the limitations are derived from the difficulty to clearly identify the areas of medical research with gaps of knowledge or waste research, in order to provide information for funding agencies. Considering the broad spectrum of this objective, we were forced to limit to 20 health conditions with the highest burden, and to provide a classification for the assessed therapies and outcomes.
Finally, due to the innovative nature of their study, it would be useful to provide future considerations that would optimize their study that are not in place, for example the role of technology in gap identification and research prioritization.	Thanks for this comment. Nevertheless, we would prefer to discuss the role of technology after developing and populating the gap maps, considering that we use technology in many steps of the process.
Overall, I look forward to there study and study findings. All the best.	Thank you very much.

Reviewer 2: Prof. Lars Gustafsson, Karolinska Institutet.

Reviewer's feedback	Authors' response
The authors are presenting a protocol for a review of clinical Chilean research publications from 2000	Thank you for your comments. According to The World Bank data, Chile is a high-income country (https://data.worldbank.org/country/chile). Nevertheless, as

linked to an analysis of prioritized health research. It is a welcomed study but the the submitted study protocol manuscript needs to be revised and clarified concerning design, used methods and on the linkage to prioritized health conditions in Chile. MAJOR COMMENTS 1. I have a hard time to understand that the listed prioritized health conditions in Table I represent the ESSENTIAL HEALTH NEEDS in Chile. Chile represents to me a middle income country that certainly has to be stuck by preventable disorders like alcohol and drug abuse, road accidents and nutritional disorders among children and need to handle multidiseased patients. The authors need to convince me and the readers that the list of health conditions do represent the 20 most important according to burden of disease. Table I represents a focus on single organ based diseases that generally do not represented the highest burden of disease in most countries.	you state, conditions such as alcohol and drug abuse or nutritional disorders carry an important burden of disease for the country. The Chilean government has included 85 conditions in the General Regime of the Health Guarantees Act (Garantías Explícitas en Salud, GES). Although these conditions have an overall high burden of disease, a condition with a high burden of disease may not have been prioritised in this health policy for other reasons (e.g. road injuries or anxiety disorders). Table 1 considers the conditions with the highest burden of disease (according to Global Burden of Disease, GBD), but only among those included in GES. To clarify this, we have rephrased the following phrase in page 9: "Then, we selected the 20 conditions with the highest burden in Chile, among those included in GES." Unfortunately, GBD is the only updated source available for our country regarding the burden of disease. We have requested this information from the Chilean Ministry of Health, but there is no other updated source. Therefore, we have matched GBD conditions to health conditions included in GES. We think that providing the list of all the GES conditions with their respective DALYs is clear enough to point out the importance of the burden of these diseases.
2. It is not clear what type of publications are meant to be included in the review. Please, clarify since it is not clear why mixed-methods projects and laboratory studies are not included (page 5). New laboratory diagnostic methods might be key both in direct care and in preventive studies.	Our team of researchers discussed what types of research design we would consider as Clinical research, and although difficult to clearly limit, we finally decided that laboratory studies with no data regarding health outcomes in individual patients (e.g. studies assessing cell lines), whilst important, usually require clinical validation before its implementation. Any study assessing treatments or interventions derived from laboratory studies will also be included. After discussion, we defined clinical studies as those focused on a clinical health topic describing, measuring or exploring a health-related outcome on human participants. We rectified our inclusion criteria, incorporating mixed-methods studies for objective 1.
The authors list an authoritative number of sites to be searched for Chilean publications (top page 8) and that is adequate. However, I do not understand the criteria for hand search strategy in a number of sites including various repositories.	Thanks for the comment. We incorporated into the manuscript the full detail of all those national and regional sources that we considered relevant to find relevant grey literature.
It might be relevant to include a hand-search but in that case it IS NOT RECOMMENDED TO INCLUDE PUBLICATIONS FOR ANALYSIS UNLESS THEY HAVE UNDERGONE	We absolutely agree with the reviewer's comment, We are not including any study if it has not undergone peer review. The repositories include peer-reviewed research.

PEER-REVIEW.	
The authors are recommended to include a flow-chart Table summarizing the different types of publication sites to be searched and criteria for inclusion and exclusion of papers in the final analysis.	Thank you for noticing this. We added the following sentence in the "Study selection" subheading (page 7): "We will present a PRISMA 2020 flow diagram showing the process of study selection."
3. The authors should revise parameters to be reviewed in each paper and consider to list if the authors REPORT CONFLICT OF INTERESTS. The authors underestimate how bias and conflict of interests can affect results of various clinical studies. Recently, this issue was addressed by an excellent review in British Medical Journal related to medical product industry (Chimonas S, Mamoor M, Zimbalist SA, Barrow B, Bach PB, Korenstein D. Mapping conflict of interests: scoping review. BMJ 202;375:e066576.) It might be adequate to include references addressing the issues on how conflict of interests bias selection of research areas and study designs.	Thanks a lot for your input. We will consider the report of conflicts of interest. We have added this in 'Data extraction' section as follows: "Conflicts of interest: We will extract the authors' statements from each report, considering descriptions for each author if available." Furthermore, as we mentioned in the manuscript, we will assess the risk of bias of the different considered study designs (see Objective 3), using standardised tools (such as Cochrane risk of bias)..
4. This study protocol is designed based on the concept that it is easy to define what are the needs of health research. In fact, definition of the health and research needs need to be open for discussion eventhough if they are defined of public bodies like ministries. It seems as if the authors underestimate the power of curiosity driven basic as well as clinical research. Please, address in the background how you consider the role of basic and clinical research to address major health and medical needs in Chile. It might be a drawback for Chilean health systems to thrive in a university and university hospital with adequate resources for curiosity driven basic and clinical research.	Indeed we believe it is basic science where most understanding about mechanisms, diagnoses, and therapies in humans are first developed and vital for medicine, and it is the translation from the lab to the clinical arena what finally derives in patients' health. Nevertheless, this study has been funded by the Chilean government to specifically provide information for guiding clinical research agendas. Basic and translational research will probably be addressed in the future by the Chliean government, but this should consider the difficult tasks and big challenges faced by the basic investigators, which is not the main expertise of this research team.
5. English grammar and style need to carefully reviewed.	Many thanks for this comment. We have detected and corrected some typographical/grammatical errors.
MINOR ISSUES 6. The listed key words on the front page do not seem adequate for the title of the paper and do not agree	We have modified the list of keywords: "Keywords: clinical research, evidence map, scoping review, Chile"

with the key words listed on page 2.	
7. Strengths and limitations of the study-page 2-3 should be revised and not only describe what the study is about.	Thank you for your feedback. We have revised the strengths and limitations, and modified it as follows: “Strengths and limitations of this study: We aim to comprehensively search, identify and classify Chilean clinical research studies by conducting a broad electronic and hand search strategy. We will generate and populate evidence gap maps for the 20 most burdensome conditions in the Chilean General Regime of the Health Guarantees Act. We will limit the evidence gap maps only to health interventions in prioritised outcomes. By focusing on the 20 conditions with the highest burden in Chile, we will not address the whole extent of the locally-prioritised health conditions. This project will guide local stakeholders and funding agencies on the identification of current knowledge gaps and redundant research, and the establishment of future research agendas.”
8. A number of terms are used in the article. Consider to include a list of definitions. Clarify also what you mean of waste of research. You cite a number articles on this subject but the term waste is not well-defined.	We changed the term “waste research” for the term “redundant research”, which is the type of waste research that we will address in this study. We believe this is a more specific term for this case. We have amended and replaced the term along the manuscript.
9. Please clarify how the study is funded and the role of funding agencies.	This study was funded by the Chilean Government through the Chilean National Agency of Research and Development (ANID). The funder has no role during the conduction of the research nor the results, and their only role is transfer of state funds to the universities involved. We have added this phrase in the page 15: “Funding: FONDECYT Grant 1212037 from the Chilean National Agency of Research and Development (ANID). The funding agency had no involvement in the conception, development, drafting or approval of this manuscript.”
10. It is remarkable that the authors do not state if you have any conflicts of interests.	Thank you for your comment. We have discussed our own conflicts of interest, and rephrased this section as follows: “Competing interests: All the authors have conducted clinical research in Chile or associated with Chilean researchers or institutions. Nevertheless, decisions about inclusion, data extraction and quality assessment of these studies, will be conducted by independent researchers within our team. We declare no other conflict of interests.”
11. I can not find that the authors have completed the PRISMA protocol as requested by the journal.	We have completed and uploaded the PRISMA-P checklist as requested.

VERSION 2 – REVIEW

REVIEWER	Gustafsson, Lars Karolinska Institutet, Department of Laboratory medicine
REVIEW RETURNED	28-Feb-2022

GENERAL COMMENTS	The manuscript is revised to a minor extent as compared to the original version despite numerous comments and questions. Please, carefully consider again my detailed comments. Issues that need to be addressed:  1. The key words to not correctly define this protocol as a bibliometric study on published Chilen health studies in specified organ based areas. Please, correct. 2. I can not find that strengths and limitations of the study is anything more than a description of the methods of the study. There are contents and methods based limitations that need to be addressed in this section of the manuscript. 3. It is unclear to me what type of studies are to be included. The authors have answered that only peer review papers are to be published. Still a number of publications that seem to be project like reports need to be included according to the Methods section. 4. Why could not you review complex multicomponent diseases that most likely need to be reviewed in order to understand burden of disease in Chile. It is not recommended to pursue a study based on administrative definitions by the management of healthcare in Chile. 5. Please, consider to simplify and shorten the manuscript in order to make it easier for the reader. Preferably Table I can be 'moved to the Appendix section.
---

VERSION 2 – AUTHOR RESPONSE

Reviewer 2: Prof. Lars Gustafsson, Karolinska Institutet.

Reviewer's feedback (sic)	Authors' response
The key words to not correctly define this protocol as a bibliometric study on published Chilen health studies in specified organ based areas. Please, correct.	Thank you for your comment. We have added the corresponding keyword. We trust we have adequately accomplished your suggestion.
I can not find that strengths and limitations of the study is anything more than a description of the methods of the study. There are contents and methods based limitations that need to be addressed in this section of the manuscript.	Thanks for your input. We have added a discussion section at the end of the manuscript, addressing your comment.
It is unclear to me what type of studies	Many thanks for noticing this. At the final paragraph of

are to be included. The authors have answered that only peer review papers are to be published. Still a number of publications that seem to be project-like reports need to be included according to the Methods section.	sub-section 'Eligibility criteria' we specified that we will exclude preprint reports with no peer review process. Regarding the 11 repositories mentioned in our hand search strategy, we will only retrieve articles published in not indexed (in electronic databases) local peer reviewed journals. Thus, in the second paragraph of the sub-section 'Hand search strategy', we have clarified this point as follows: «We identified seven information resources considered relevant to identify all the evidence that has not been indexed in the databases described in the electronic search and additional four information resources necessary to retrieve trials, reviews, protocols or other types of evidence, published in peer reviewed journals, that contain the inclusion criteria.»
Why could not you review complex multicomponent diseases that most likely need to be reviewed in order to understand the burden of disease in Chile. It is not recommended to pursue a study based on administrative definitions by the management of healthcare in Chile.	Thank you for your feedback. We have added this as a limitation in our discussion. However, we believe that any prioritisation of this kind must be conceptualised considering local factors, and the organic emergency of specific needs or social value for each context (as defined by the policy-makers or other stakeholders, for instance), instead of a systematic approach.
Please, consider to simplify and shorten the manuscript in order to make it easier for the reader. Preferably Table I can be moved to the Appendix section.	We thank you for this commentary. Our protocol is quite exhaustive and largely detailed. In this manuscript, we aim to display a specific roadmap of what will be our four-years project (funded by a national agency), explaining each step of our procedures. We have attempted to simplify this as much as we considered adequate, according to our criteria and the instructions for authors of BMJ Open. We believe that it is important to provide Table 1 within the manuscript, because it shows the data that feeds the subsequent analysis. However, we will be looking forward (and be able to kindly accept) any additional editorial requirement on this matter.

VERSION 3 – REVIEW

REVIEWER	Gustafsson, Lars Karolinska Institutet, Department of Laboratory medicine
REVIEW RETURNED	26-Apr-2022
GENERAL COMMENTS	The authors have made minor changes of the manuscript according to suggestions but have chosen to keep the length of the manuscript. It is still recommended to shorten the manuscript. Please, add bibliometric study to the key words.

VERSION 3 – AUTHOR RESPONSE

We have shortened the length of our article, and moved the list of data sources for the hand search strategy to the appendices. We have also added bibliometrics as a keyword.